# Vaccine co-display of CSP and Pfs230 on liposomes targeting two *Plasmodium falciparum* differentiation stages

Wei-Chiao Huang[1], Moustafa T. Mabrouk[1], Luwen Zhou[2], Minami Baba[3], Mayumi Tachibana[3], Motomi Torii[3], Eizo Takashima[4], Emily Locke[5], Jordan Plieskatt[5], C. Richter King[5], Camila H. Coelho[6], Patrick E. Duffy [6], Carole Long[2], Takafumi Tsuboi[4], Kazutoyo Miura [2], Yimin Wu[5], Tomoko Ishino [3,7] & Jonathan F. Lovell [1]

A vaccine targeting multiple stages of the *Plasmodium falciparum* parasite life cycle is desirable. The sporozoite surface Circumsporozoite Protein (CSP) is the target of leading anti-infective *P. falciparum* pre-erythrocytic vaccines. Pfs230, a sexual-stage *P. falciparum* surface protein, is currently in trials as the basis for a transmission-blocking vaccine, which inhibits parasite development in the mosquito vector. Here, recombinant full-length CSP and a Pfs230 fragment (Pfs230D1+) are co-displayed on immunogenic liposomes to induce immunity against both infection and transmission. Liposomes contain cobalt-porphyrin phospholipid (CoPoP), monophosphoryl lipid A and QS-21, and rapidly bind His-tagged CSP and Pfs230D1+ upon admixture to form bivalent particles that maintain reactivity with conformational monoclonal antibodies. Use of multicolor fluorophore-labeled antigens reveals liposome binding upon admixture, stability in serum and enhanced uptake in murine macrophages in vitro. Bivalent liposomes induce humoral and cellular responses against both CSP and Pfs230D1+. Vaccine-induced antibodies reduce parasite numbers in mosquito midguts in a standard membrane feeding assay. Mice immunized with liposome-displayed antigens or that passively receive antibodies from immunized rabbits have reduced parasite liver burden following challenge with transgenic sporozoites expressing *P. falciparum* CSP.

[1] Department of Biomedical Engineering, University at Buffalo, State University of New York, Buffalo, NY 14260, USA. [2] Laboratory of Malaria and Vector Research, National Institute of Allergy and Infectious Diseases, National Institutes of Health, Rockville, MD 20852, USA. [3] Division of Molecular Parasitology, Proteo-Science Center, Ehime University, Toon, Ehime 791-0295, Japan. [4] Division of Malaria Research, Proteo-Science Center, Ehime University, Matsuyama, Ehime 790-8577, Japan. [5] PATH's Malaria Vaccine Initiative (MVI), Washington, DC 20001, USA. [6] Laboratory of Malaria Immunology and Vaccinology, National Institute of Allergy and Infectious Diseases, National Institutes of Health, Rockville, MD 20852, USA. [7] Department of Parasitology and Tropical Medicine, Tokyo Medical and Dental University, Bunkyo-ku, Tokyo 113-8510, Japan. ✉email: tishino.vip@tmd.ac.jp; jflovell@buffalo.edu

Malaria is caused by *Plasmodium* parasites in a continuous cycle of mosquito-to-human and human-to-mosquito transmission. Despite progress in controlling the disease, hundreds of millions of cases still occur annually, causing over 627,000 deaths[1]. Incidence actually increased somewhat from 2015 to 2019, with the highest caseload in sub-Saharan Africa, and with children under five comprising two-thirds of fatalities[2]. The development and distribution of an effective malaria vaccine is likely required to eliminate the global burden of the disease.

The Circumsporozoite Protein (CSP) is localized on the surface of sporozoites, which migrate to infect hepatocytes after deposition in the skin by mosquito bites, and it plays important roles in sporozoite formation, migration and invasion[3,4]. Antibodies against CSP can prevent sporozoite migration and infection of hepatocytes in vitro and in vivo[5]. The CSP protein sequence (NF54 allele) consists of an immunodominant central repeat region, with a junctional "NPDP" sequence, 38 "NANP" copies, and four "NVDP" copies[6] that all serve as B-cell epitopes for human monoclonal antibodies with high binding affinity[7]. Several human T cell epitopes have been identified within CSP[8–10].

To date, the leading malaria vaccination strategy has focused on the pre-erythrocytic parasite stage; preventing infection after the mosquito spreads parasites to a human. The RTS,S antigen comprises the hepatitis B surface antigen fused to a truncated CSP with 18 NANP repeats and the C-terminus up through a portion of the glycosylphosphatidylinositol (GPI) anchor addition sequence, which self-assembles into virus-like particles. The Mosquirix® vaccine (RTS,S/AS01) was recently approved by the World Health Organization (WHO) for use in children at risk in areas of moderate to high *P. falciparum* malaria transmission, based on data from a large Phase 3 clinical trial and a pilot implementation program in three sub-Saharan African countries[11,12]. However, the efficacy of RTS,S/AS01 declines over time, justifying the search for improved anti-malaria vaccines[13,14]. A Phase 2b trial of R21/Matrix-M, another CSP-based vaccine (that features reduced content of the hepatitis B surface antigen scaffold protein) recently demonstrated promising efficacy in clinical testing[15]. A Phase 1 dose-escalation clinical trial of full-length CSP admixed with or without a synthetic monophosphoryl lipid A (MPLA) and *Quillaja saponaria* 21 (QS-21) liposome adjuvant was shown to be well-tolerated and immunogenic[16]. Another study in non-human primates showed that the CSP antigen formulated with the Army Liposomal Formulation containing MPLA and QS-21 had strong immunogenicity[17].

Transmission-blocking vaccines (TBVs) aim to induce human antibodies that impair the development of sexual stage parasites, and vaccine activity is measured by assessing the reduction in oocyst counts on mosquito midguts after a blood meal containing gametocytes and vaccine-induced antibodies. TBVs reduce or block downstream sporozoite invasion in the mosquito salivary gland and are thereby expected to reduce transmission of the parasite and incidence of disease at the community level provided sufficiently high activity and vaccine coverage. Numerous TBV antigens have been the focus of vaccine research efforts, including Pf48/45, Pfs25, Pfs28, PfHAP2 and Pfs230[18–20]. Pfs230 is a large protein containing over 3,000 amino acids (aa), fragments of which have been assessed as candidate TBV antigens[21]. A recent clinical trial demonstrated that Pfs230D1M (Pfs230 aa range 542-736; NF54 allele) prepared as a protein-protein conjugate vaccine induces serum transmission reducing activity (TRA) in humans[22].

Vaccines containing multiple antigens from different malaria life stages have been proposed as a strategy to improve efficacy[23]. Several multi-stage malaria vaccine candidates have progressed to early phase clinical trials but so far none have demonstrated immune responses that led to further development[24–26].

Preclinical research efforts have examined multi-stage antigen constructs for malaria including Pfs25 and CSP using fusion antigens displayed on a viral vector[27], and Pfs25 and *P. falciparum* merozoite surface protein 1 (MSP1) admixed with an emulsion of synthetic MPLA in a stable squalene-in-water emulsion[28]. Further efforts have also sought to combine multiple target antigens in a single molecule, including ProC6C which combines the Pfs230, Pfs48/45 and CSP molecules into a single antigen, and in mice demonstrated the ability to elicit antibodies that inhibited both transmission and sporozoite invasion when formulated in the Saponin based adjuvant Matrix-M™[29].

In the present study, we co-display *P. falciparum* TBV and pre-erythrocytic antigens on the surfaces of immunostimulatory liposomes. The TBV antigen, Pfs230D1+ (abbreviated as "230" herein in figures and in some descriptions for simplicity), comprises residues 552–731 of Pfs230 (NF54 allele) and was selected for optimized expression of an intact, nonglycosylated, properly folded immunogen in the baculovirus expression system[30]. This amino acid also avoids an O-linked glycosylation site and corresponds to the C-terminus of the prodomain and the first cysteine motif domain of the protein. The pre-erythrocytic antigen CSP was produced as full-length (aa 27-383; NF54 allele) with full repeat region (4 NVDP and 38 NANP repeats) in the *L. lactis* system as previously described[6]. We make use of cobalt-porphyrin-phospholipid (CoPoP) liposome technology for non-covalent display of antigens on liposome surfaces, which has been shown to be an effective approach for TBV antigens with simple admixture[31–33]. Antigen binding and conformational integrity on CoPoP liposomes are assessed, as is immunogenicity in mice and rabbits. Overall, we find that liposome-displayed bivalent antigens are easily assembled and effectively induce functional immune responses against both CSP and Pfs230D1+.

## Results

**Characterization of dual-stage antigen binding to CoPoP liposomes.** The concept of bivalent antigen display using CoPoP liposomes is shown in Fig. 1a. Upon admixture with the soluble and previously-characterized Pfs230D1+[30] and CSP antigens[6], their C-terminus His-tag inserts into the bilayer of liposomes containing CoPoP, based on intrabilayer cobalt interaction with the imidazole group of the histidine residues[34]. The amino acid sequences of Pfs230D1 and CSP used in this study are shown in Supplementary Table S1. In addition, the synthetic MPLA analog PHAD-3D6A (abbreviated PHAD herein) and QS-21 were included in the bilayer, since MPLA and QS-21 have been shown to be synergistic immune stimulating adjuvants in the potent AS01 human vaccine adjuvant[35]. The ratio of active components in the liposomes was previously optimized with a mass ratio of [dioleoylphosphatidylcholine (DOPC): cholesterol: CoPoP: PHAD: QS-21] as [20: 5: 1: 0.4: 0.4] to generate the "CPQ" adjuvant system[36,37]. Analogous liposomes containing CoPoP and PHAD, but lacking QS-21 are termed "CP" herein. To display antigens on liposomes, CP was simply admixed with CSP and Pfs230D1+. Since the molecular weight of CSP (~39 kDa) was nearly twice that of Pfs230D1+ (~21 kDa), the CSP mass concentration was doubled, in order to provide roughly equimolar components of the antigens. Figure 1b (and full gel shown in Supplementary Fig. S1) shows that as the liposome to duplexed protein ratio is increased, CSP and Pfs230D1+ were increasingly converted into liposome-displayed format forming either individual antigen particles displaying CSP or Pfs230D1+, or bivalent antigen particles displaying both antigens. To assess particle formation, the assembled formulations were incubated with Nickel-nitrilotriacetic acid (Ni-NTA) magnetic beads, which bind freely soluble, but not liposome-bound antigens. These beads can

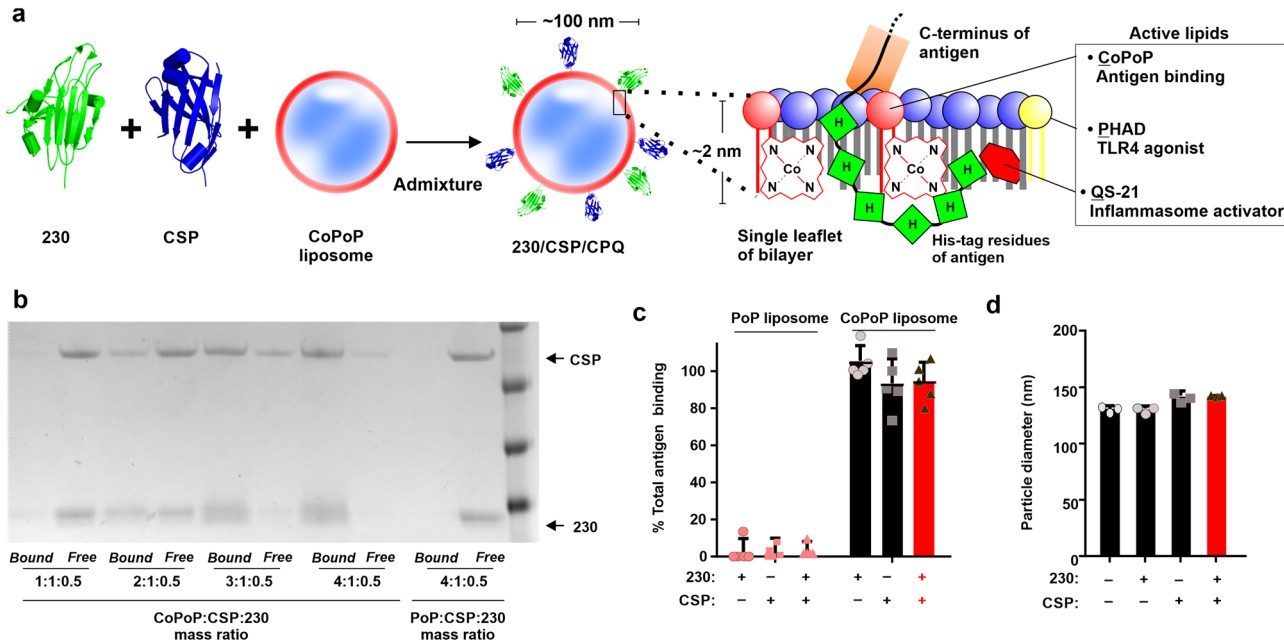

**Fig. 1 Bivalent particle formation upon admixture of CSP and Pfs230D1+ with CoPoP liposomes. a** Schematic representation of the bivalent CoPoP liposomes displaying Pfs230D1+ ("230") and full-length *P. falciparum* CSP ("CSP"). **b** Antigen binding to CP liposomes was measured by Ni-NTA bead competition assay. Identical PoP liposomes lacking cobalt served as a control. **c** Binding percentage of antigens to CoPoP or PoP liposomes using a mass ratio of [CoPoP or PoP: CSP: 230] = 4:1:0.5. **d** Particle size before and after antigen binding. Bars show mean ± s.d for n = 3-5 replicates. Red bars indicate bivalent, liposome-displayed CSP/230.

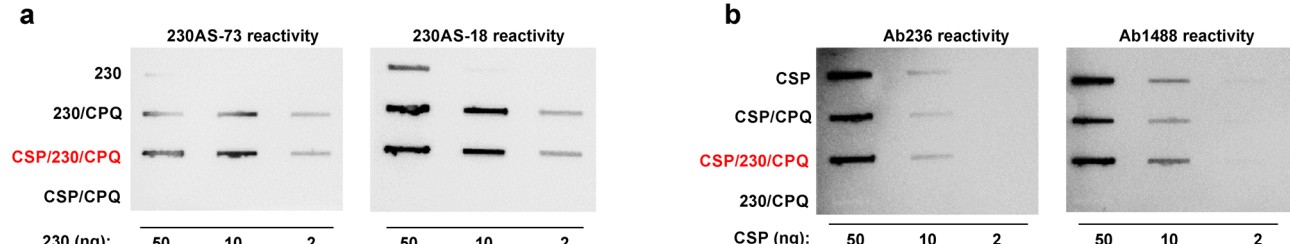

**Fig. 2 Liposome-displayed malaria antigens maintain conformational integrity.** Slot blot detection of mAb binding to adsorbed Pfs230D1+ and CSP in soluble or particulate form. Human monoclonal antibodies that target (**a**) Pfs230 (230AS-73 and 230AS-18) or (**b**) CSP (Ab236 and Ab1488) were incubated with the membrane and bound mAbs were detected using a chemiluminescent secondary antibody. Red text indicates bivalent, liposome-displayed CSP/230.

rapidly be separated and protein content in the beads (soluble protein) or supernatant (liposome-bound protein) is determined. A [4: 1: 0.5] mass ratio of [CoPoP: CSP: 230] resulted in complete conversion of both antigens into liposome-bound format. This ratio of CoPoP to antigens was used in further studies. In contrast, liposomes that had identical lipid content, but lacked cobalt in the CoPoP lipid (termed "PoP liposomes") exhibited virtually no protein binding. Antigen binding was also confirmed with an independent high-speed centrifugation assay, which results in the pelleting of liposomes and any bound protein (Fig. 1c). After incubation with CoPoP liposomes, more than 80% of the total protein bound to the liposomes in individual or duplexed format. Antigens did not bind to cobalt-free PoP liposomes. The size of the liposomes was measured before and after incorporation with antigens, as shown in Fig. 1d. CoPoP liposomes had a diameter between 125 and 150 nm, and remained stable after binding the individual or the bivalent antigens, as evidenced by the lack of major increase in liposome size.

The conformational integrity of Pfs230D1+ and CSP in liposome-bound format was assessed using a slot blot assay. Conformational monoclonal antibodies (mAbs) that target

Pfs230D1+ (Fig. 2a) or CSP[38] (Fig. 2b) were reacted with antigens in soluble or liposome-displayed format, following antigen adsorption to a nitrocellulose membrane. The 230AS-73 and 230AS-18 mAbs recognize Pfs230D1+ in non-reducing but not in reducing conditions, as indicated by Western blot (Supplementary Fig. S2). Based on qualitative slot blot analysis, both mAbs recognized Pfs230D1+ in liposome-bound form regardless of whether it was displayed alone or in bivalent format. Interestingly, 230AS-73 and 230AS-18 recognized soluble Pfs230D1+ with diminished efficiency compared to liposome-bound protein, which could be related to adsorption efficiency of the soluble small protein on the membrane during the slot blot, a phenomenon that has been observed previously[39]. As expected, liposomes displaying CSP were not recognized by the Pfs230 mAbs. On the other hand, the mAbs Ab236 (a CSP C-terminal region–specific mAb) and Ab1488 (another CSP C-terminal region–specific mAb) recognized CSP in both soluble form and liposome-displayed form[40], indicating that the antigens retained epitope accessibility. Altogether, these qualitative results provide direct evidence that the conformation of antigens displayed on CoPoP liposome remained intact.

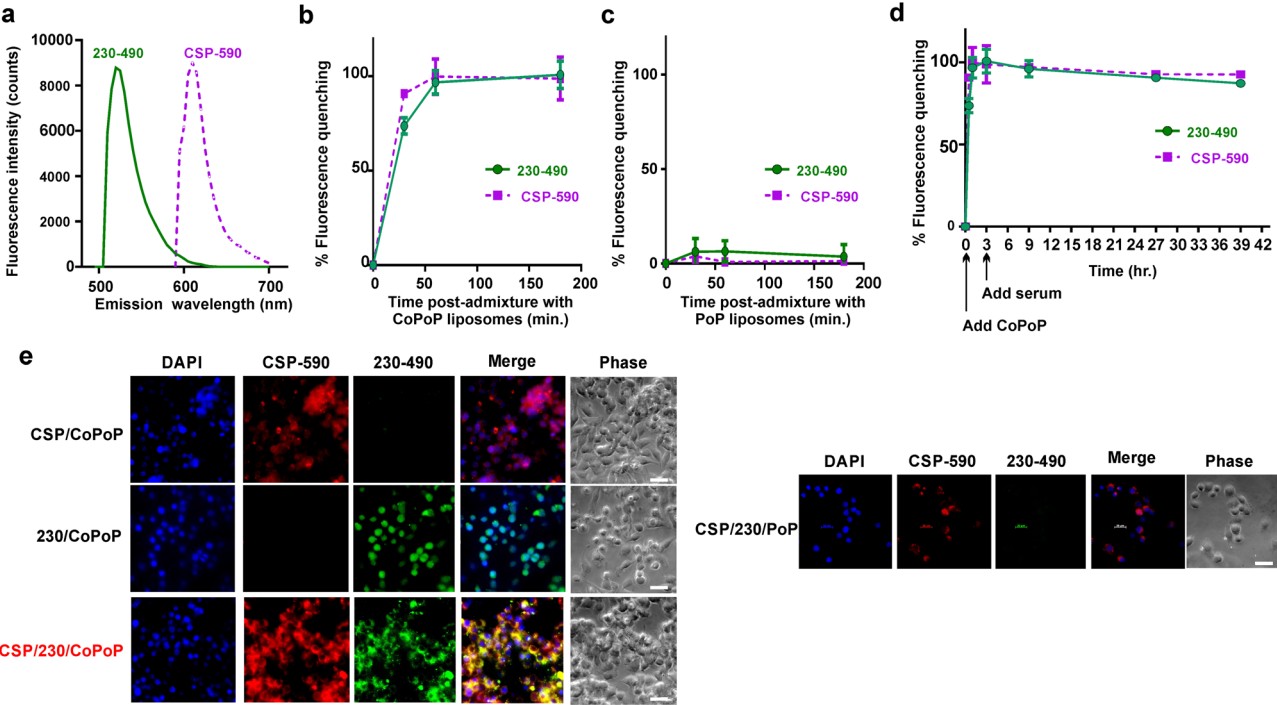

**Fig. 3 Fluorescently-labeled CSP and Pfs230D1+ reveal particle formation, stability and macrophage uptake. a** Fluorescence emission spectra of fluorophore-labeled CSP ("CSP-590") and Pfs230D1+ ("230-490"). Binding kinetics of the bivalent antigens, measured in a single assay in real-time, as reflected by fluorescence quenching upon admixture with (**b**) CoPoP liposomes or (**c**) PoP liposomes, which lack cobalt in the bilayer. **d** Serum stability of liposome-displayed bivalent antigens. Labeled antigens were first incubated with CoPoP liposomes for 3 h, then human serum was added where indicated to a final concentration of 20 % (v/v). **e** RAW264.7 cell uptake of bivalent antigens admixed with CoPoP liposomes and PoP liposomes for 4 h. Bar: 2 μm. Graph points show mean ± s.d. for $n = 3$ replicates.

Next, the particle-formation kinetics and serum stability of duplexed, liposome-displayed antigens was investigated. A fluorescence resonance energy transfer (FRET) assay was developed by covalently labeling Pfs230D1+ and CSP with spectrally distinct small molecule FRET donor fluorophores to track fluorescent quenching that occurs upon binding CoPoP liposomes, which serve as a FRET acceptor. As shown in Fig. 3a, labeled CSP ("CSP-590") had minimal spectral emission overlap with labeled Pfs230D1+ ("230-490") The binding of both antigens to CoPoP (Fig. 3b) or otherwise identical but cobalt-free PoP (Fig. 3c) liposomes could then be simultaneously monitored in real time, as indicated by fluorescence quenching. The binding of CSP-590 and 230-490 to CoPoP liposomes proceeded rapidly at room temperature, with the process nearly completing within an hour. No significant binding was observed between the antigens and PoP liposomes lacking cobalt. The same FRET assay was used to gauge the stability of the liposome-bound antigens in the presence of sera (Fig. 3d). Labeled CSP and Pfs230D1+ were first incubated with liposomes, which resulted in rapid quenching of their fluorescence emission. Human serum was then added to a final concentration of 20% and the liposomes were incubated at 37 °C for up to 36 h. During this period, the labeled antigens remained quenched, reflecting stable antigen association with CoPoP liposomes. The use of the fluorescently labeled antigens also permitted observation of antigen uptake in murine RAW264.7 macrophages in vitro. As shown in Fig. 3e, when liposome-bound antigens were incubated with macrophages individually or in bivalent format, strong uptake in macrophages cells was apparent, based on observation by fluorescence microscopy. Since the antigens are quenched in their bound form, these data also suggest that upon uptake, the antigens and liposomes dissociate, at least in part, likely due to proteolytic degradation or the altered biochemical environment of endocytic

vesicles. When the bivalent antigens were admixed with analogous PoP liposomes lacking cobalt, then incubated with macrophages, minimal antigen uptake was observed. Thus, liposome-bound antigens were better delivered to antigen-presenting cells in vitro.

**Antibody responses from bivalent, liposome-displayed CSP and Pfs230D1+ in outbred mice.** CSP and Pfs230D1+ were used to immunize outbred ICR mice intramuscularly on day 0 and 21, with sera collection on day 42. Based on ELISA analysis, bivalent, liposome-displayed antigens ("CSP/230/CP") induced stronger antigen-specific IgG titers compared to mice immunized in the same conditions but with admixture with Alhydrogel (Alum), for both CSP (Fig. 4a) and Pfs230D1+ (Fig. 4b). Monovalent liposome vaccines did not induce appreciable antibodies against the CSP or Pfs230D1+ antigens that were not included in the formulation. The bivalent CSP/230/CP group received a half of the dose of individual immunogen compared to monovalent CSP/CP or 230/CP groups, and showed trending lower ELISA units. However, the differences were not statistically significant (Supplementary Table S2).

To assess the transmission reducing activity of the induced Pfs230D1+ antibodies, the standard membrane feeding assay (SMFA) was employed. SMFA results demonstrated that admixture of bivalent antigens with CP was more effective than Alum in inducing antibodies that reduced parasite transmission in mosquito midguts. As shown in Fig. 4c, more than 95% inhibition in oocyst density was observed with 750 μg/ml purified IgG following 230/CP or CSP/230/CP immunization. The antibodies induced by Alum had no functional activity in the SMFA. The raw data are included in Supplementary Fig. S3a and Supplementary Table S3 (**SMFA# 276-1**). Post-immune sera showed reactivity with sporozoites and gametocytes in an immunofluorescence

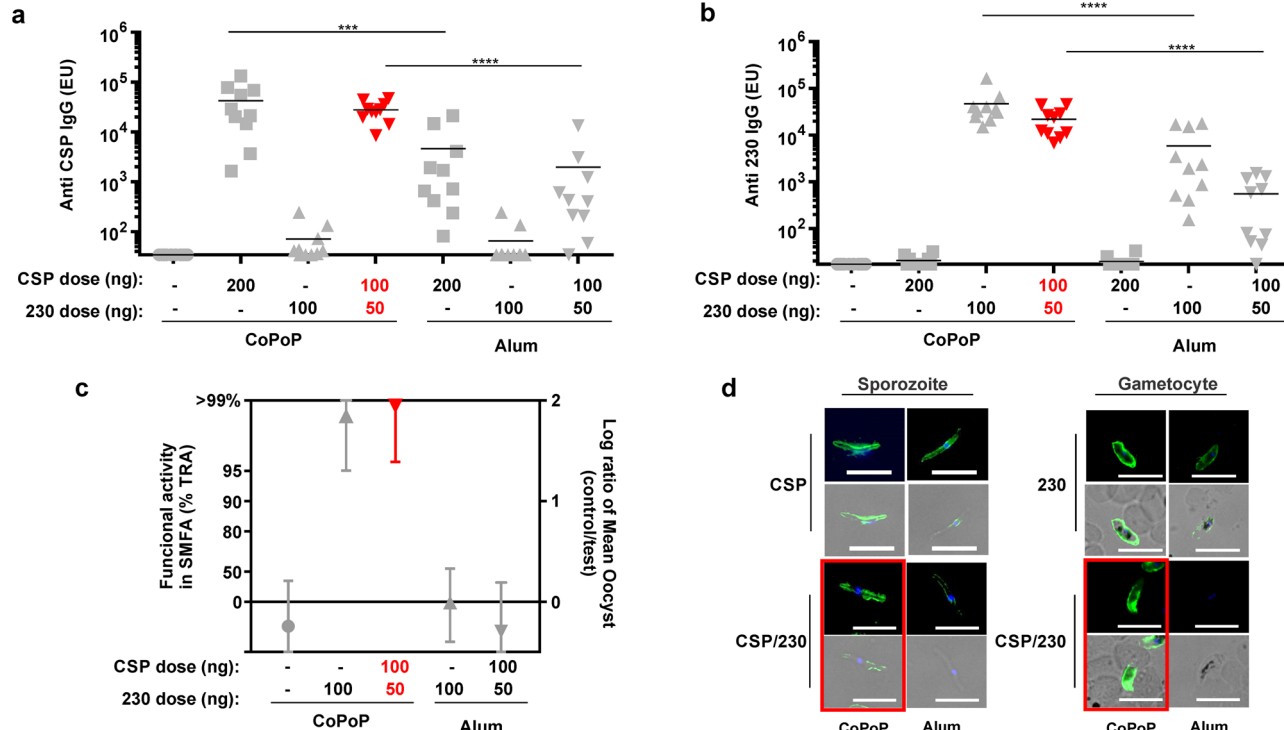

**Fig. 4 Bivalent CSP and Pfs230D1+ immunization of outbred mice.** ICR mice were immunized with CSP, Pfs230D1+ or bivalent antigens with CoPoP liposomes or alum on day 0 and day 21, final bleeding were collected on day 42. **a** Anti CSP ELISA units (EU) and (**b**) Anti Pfs230D1+ EU. ELISA experiments were performed with n = 10 mice. For A and B, bars represent geometric mean, and one-way ANOVA followed by Tukey test using log-transformed data were used to compared difference among CSP/CP, CSP/230/CP, CSP/Alum and CSP/230D1/Alum for **a** and 230/CP, CSP/230/CP, 230/Alum and CSP/230/Alum for **b**. ****$p < 0.0001$, ***$p < 0.001$, **$p < 0.005$. **c** Standard membrane feeding assay (SMFA) results of purified IgGs tested at 750 µg/ml with complement. %TRA and the 95% confidence interval (95% CI) are shown. **d** Immunofluorescence assay of fixed NF54 parasites at sporozoite and gametocyte stages. Bar: 10 µm. Image was taken from a single experiment. Red symbols or outlines indicate bivalent, liposome-displayed CSP/230.

assay (fixed IFA), indicating the presence of antibodies capable of recognizing native CSP and Pfs230, respectively (Fig. 4d).

A second immunization study was conducted in outbred mice using CP liposomes that also included QS-21, referred to as CPQ. In this experiment, the outbred mice were immunized with the same doses of each immunogen for monovalent and bivalent groups, ranging from 0.25–2 µg antigen. Statistically significant interferences were not observed in antibody titers (Supplementary Fig. S4a, b) and SMFA activity (Supplementary Fig. S4c). In addition, immune interference was also assessed for cellular immunity. CD8+ T cells were identified by gating in TCRβ+CD8+ T cells, and examining CD44hi population, followed by gating population of IL2+TNFa+IFN-γ+ cells (an example of gating strategy is seen in Supplementary Fig. S5) and CD4+ T cells were identified by gating in TCRβ+CD4+ T cells, and examining Foxp3 negative then gated with CD44hi population, followed by gating population of IL2+TNFa+IFN-γ+ cells (Supplementary Fig. S6). Such antigen-specific, triple cytokine-producing polyfunctional CD4[41] and CD8 T[42] cells are of interest for cell-mediated malaria immunity. After stimulation with the indicated antigen, higher frequencies of triple-cytokine-expressing CD8+ and CD4+ T cells were observed in mice immunized with bivalent antigen admixed with CPQ compared to the control group, and no distinct immune interference was detected (Supplementary Fig. S7A–D).

**Humoral and cellular responses induced by bivalent, liposome-displayed CSP and Pfs230D1+ in inbred mice.** *P. berghei* sporozoites infect inbred C57BL/6 mice ~100-fold more efficiently than outbred ICR mice[43]. Transgenic rodent malaria parasites (*P. berghei*

ANKA) which express PfCSP instead of PbCSP (the PbPfCSP transgenic parasite) have been used as a model to determine the effects of anti-CSP immunity on sporozoite liver infection in C57BL/6 mice[44]. Therefore, vaccination was next assessed in C57BL/6 mice prior to PbPfCSP sporozoite challenge. Mice were immunized with 150 ng of bivalent antigens (100 ng CSP and 50 ng Pfs230D1 +) admixed with CoPoP liposomes or other comparator adjuvants. The adjuvants used include CoPoP-based CP, and CPQ, liposomal PHAD/QS-21 lacking CoPoP ("LPQ"), PoP/PHAD liposomes (2HP), and Alum. CoPoP-based formulations induced higher levels of anti-CSP (Fig. 5a) and anti-Pfs230D1+ (Fig. 5b) IgG compared to the monomer antigens (i.e., all other adjuvants). This underscores the importance of particle formation for inducing strong antibody responses. The SMFA showed that antibodies induced by CoPoP liposomes were more effective in reducing parasite transmission as shown in Fig. 5c (additional raw data is included in Supplementary Fig. S8, and Supplementary Table S3, **SMFA# 313**). The antibody levels elicited by the CPQ adjuvant formulation (which includes QS-21) were not statistically different from the CP formulation based on one-way ANOVA followed by Tukey's test (Supplementary Table S4). Nevertheless, QS-21 was included for further studies as it has been shown to enhance cellular immunity, which may contribute to protection in mouse challenge models of malaria[45]. CSP/230/CP or CSP/230/CPQ post-immunization sera from C57BL/6 mice was reactive with sporozoites and gametocytes (Fig. 5d) using an IFA. On the other hand, mice immunized with antigens admixed with formulations omitting CoPoP did not induce antibodies that could be readily detected by reactivity to sporozoites or gametocytes.

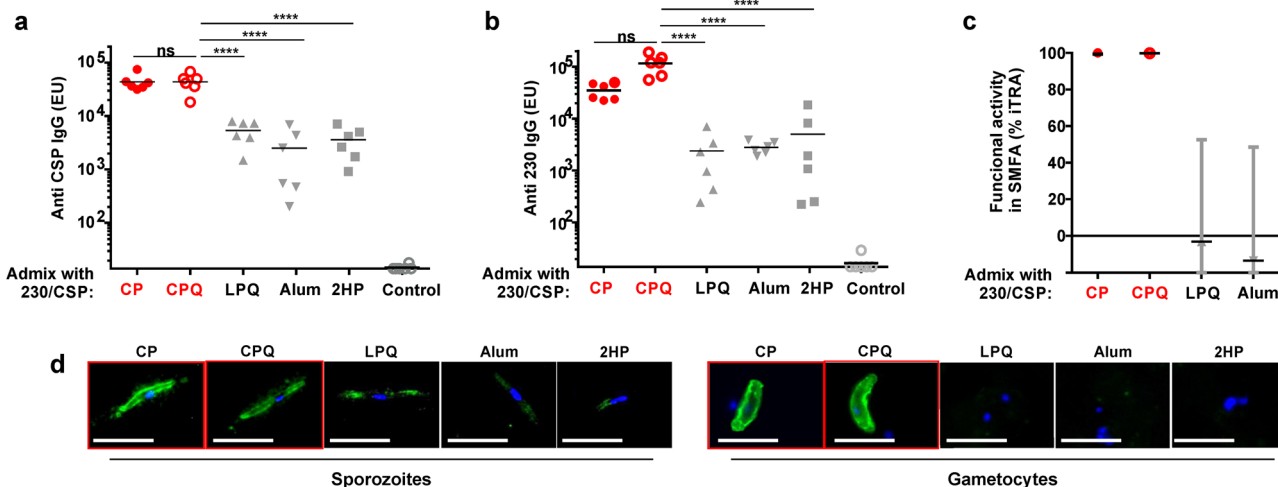

**Fig. 5 Bivalent CSP and Pfs230D1+ immunization of inbred mice.** C57BL/6 mice ($n = 6$ per group) were immunized with 100 ng CSP and 50 ng 230, admixed with indicated adjuvants on day 0 and 21. Sera were collected on day 42, and anti-CSP (**a**) and anti-230D1 (**b**) IgG levels were assessed by ELISA. One-way ANOVA followed by Tukey test using log-transformed data was used to compare statistical differences for the CSP/230/CPQ group versus other groups. ****$p < 0.0001$. Full statistical analysis is reported in Supplementary Table S4. Bars represent geometric means. **c**) SMFA results of purified IgGs tested at 750 μg/ml with complement. % Inhibition (TRA) and 95% confidence intervals are shown. **d** Immunofluorescence assay of NF54 parasites at sporozoite and gametocyte stages. Bar: 10 μm. Red symbols or outlines indicate bivalent, liposome-displayed CSP/230.

To investigate cellular responses, IFN-γ secretion of spleno-cytes from immunized mice was measured with antigen restimulation. Figure 6a shows that there was higher secretion of IFN-γ in response to Pfs230D1+ restimulation in mice immunized with CSP/230/CPQ, relative to the same bivalent antigens admixed with all other adjuvants. Based on intracellular staining, CSP/230/CPQ splenocytes restimulated with Pfs230D1+ had a higher frequency of triple-cytokine polyfunctional CD8+ T cells compared to mice immunized with the bivalent antigens admixed with 2HP or Alum, or blank CPQ control (Fig. 6b). For 230-specific polyfunctional CD4+ T cells, the CSP/230/CPQ group had the highest proportion relative to every other experimental group, including CSP/230/CP, that lacked QS-21 (Fig. 6c). The CSP-specific cellular response generally mirrored the 230 response. Splenocytes from the CSP/230/CPQ group restimulated with CSP induced higher levels of IFN-γ (Fig. 6d), and had higher frequency of polyfunctional CD8+ (Fig. 6e) and CD4+ (Fig. 6f) T cells, compared to other immunization groups. Interestingly, the absolute magnitude of the 230-specific cellular response was significantly greater than the CSP-specific cellular response, despite using only half the mass of Pfs230D1+ relative to CSP.

**Bivalent, liposome-displayed CSP and Pfs230D1+ induce protective anti-CSP immune responses.** Next, we assessed whether CSP and Pfs230D1+ displayed on CPQ liposomes could induce protective immunity with vaccination of C57BL/6 mice. Mice were immunized 3 times with high dose (2 μg CSP; 1 μg 230, 8 μg CoPoP and 3.2 μg QS-21) or low dose (0.5 μg CSP; 0.25 μg 230, 2 μg CoPoP and 0.8 μg QS-21) of bivalent antigens formulated with CPQ, or with high or low doses of monovalent CSP alone formulated with CPQ. As expected, CPQ liposomes induced specific anti-CSP (Fig. 7a) and anti-Pfs230D1+ anti-bodies (Fig. 7b) in both high and low dose of the dual antigen formulation. Full statistical analysis is listed in Supplementary Table S6. As shown in Fig. 7c, CSP/230/CPQ resulted in ~78% reduction in parasite liver burden following sporozoite infection for the high dose immunization group, and ~70% reduction for the low dose immunization group. Monovalent CSP/CPQ was somewhat less effective in reducing liver parasite burden by 57%

and 30% with the high and low dose, respectively. This was less than the CSP/230/CPQ bivalent vaccine, in agreement with the ELISA units against CSP (Fig. 8a). Neither 230/CPQ nor CPQ alone only induced inhibitory effect on infection. As shown in Supplementary Fig. S9, post immune sera of CSP/230/CPQ but not 230/CPQ recognized PbPfCSP sporozoites, but not *P. berghei* sporozoites lacking *P falciparum* CSP transgene expression.

**Antibody response induced by bivalent, liposome-displayed CSP and Pfs230D1+ in rabbits.** To determine whether anti-bodies against CSP were functional in the PbPfCSP model[6,46,47], New Zealand White rabbits were first immunized as a source of antibodies, for passive transfer to mice prior to challenge. Rabbits were thrice intramuscularly immunized with CSP (20 μg) and Pfs230D1+ (10 μg) admixed with CPQ or Alum adjuvants. Sera from two immunized rabbits in each group were pooled and IgG was purified. The ELISA units of those IgGs against CSP/Pfs230D1+ were 96,620/141,148 and 21,691/51,654 for CPQ and Alum groups, respectively. 10 mg purified IgGs were injected intravenously per mouse, who were 16 h later challenged intra-venously with 2000 PbPfCSP sporozoites. Sera from the IgG injected and non-injected (control) mice were collected prior to sporozoite challenge to determine serum antibody concentration. At that time, mice had significant levels of passively transferred rabbit antibodies circulating in sera against both CSP (Fig. 8a) and Pfs230D1+ (Fig. 8b). Transferred antibodies from rabbits immunized with liposome-displayed antigens led to significantly higher circulating antibodies compared to rabbits immunized with the same antigens admixed with Alum. As a negative con-trol, a rabbit was immunized with Pfs230D1+ (20 μg) alone with CPQ, IgG was purified, and the IgG was injected into mice. Following sporozoite challenge, mice that passively received antibodies from rabbits immunized with CSP/230/CPQ had a median of ~80% reduced liver parasite burden compared to non-immunized control mice (Fig. 8c). No protection was conferred from antibodies passively transferred from rabbits immunized with CSP/230/Alum. The positive control monoclonal antibody AB317, intravenously injected into mice at a dosage of 300 μg completely inhibited liver infection, as previously observed[48]. Thus, these data show that CSP and Pfs230D1+ admixed with

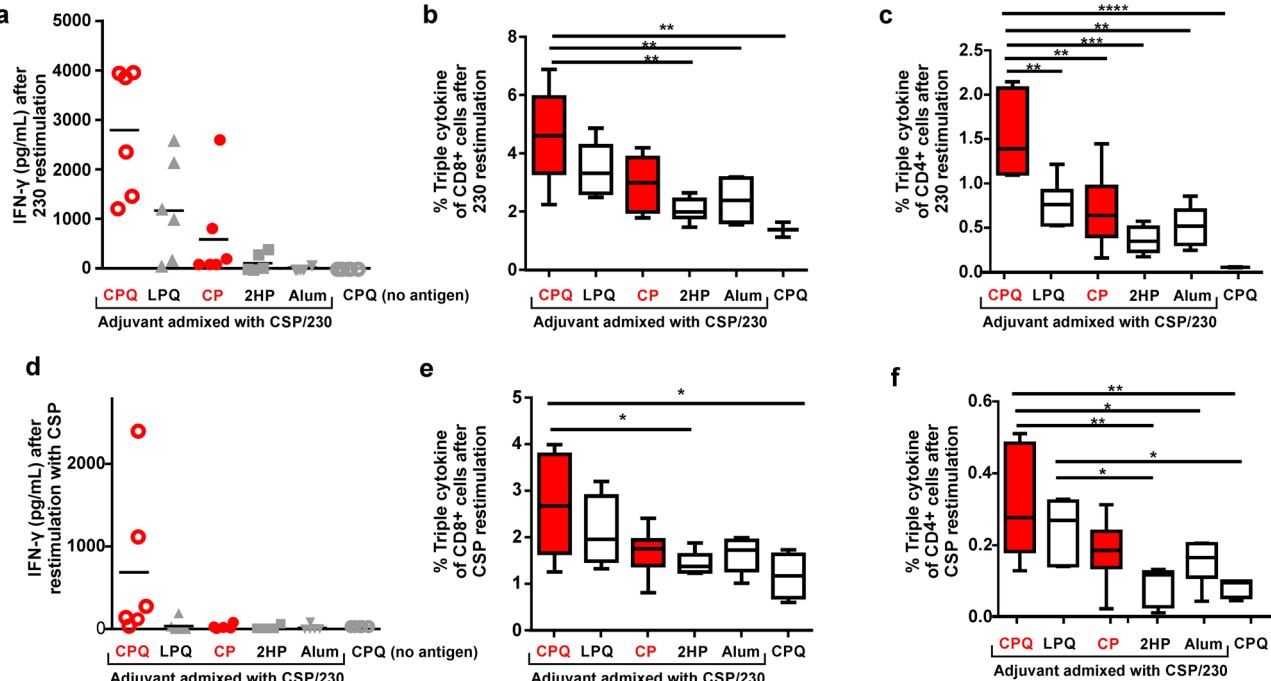

**Fig. 6 Cellular responses.** C57BL/6 mice (*n* = 6 per group) were immunized in the same manner as in Fig. 5, and on day 42, splenocytes were isolated and restimulated with 1 µg/ml CSP or 230. 230-specific cellular responses are shown with respect to (**a**) IFN-γ secretion, and IL2$^+$ TNFa$^+$IFN-γ$^+$ triple positive CD8$^+$ (**b**) and CD4$^+$ (**c**) T cells. CSP-specific cellular responses are shown with respect to (**d**) IFN-γ secretion, and IL2$^+$TNFa$^+$IFN-γ$^+$ triple positive CD8$^+$ (**e**) and CD4$^+$ (**f**) T cells. One-way ANOVA followed by Tukey test were used to compared difference. $^{****}p < 0.0001$, $^{***}p < 0.001$, $^{**}p < 0.005$, $^*p < 0.05$. For **a** and **d**, bars represent geometric means. For the box-and-whiskers plots (**b**, **c**, **e**, **f**), the line shows the median value, box shows the interquartile range and whiskers represent the data range (*n* = 6 mice per group). Red symbols or outlines indicate bivalent, liposome-displayed CSP/230.

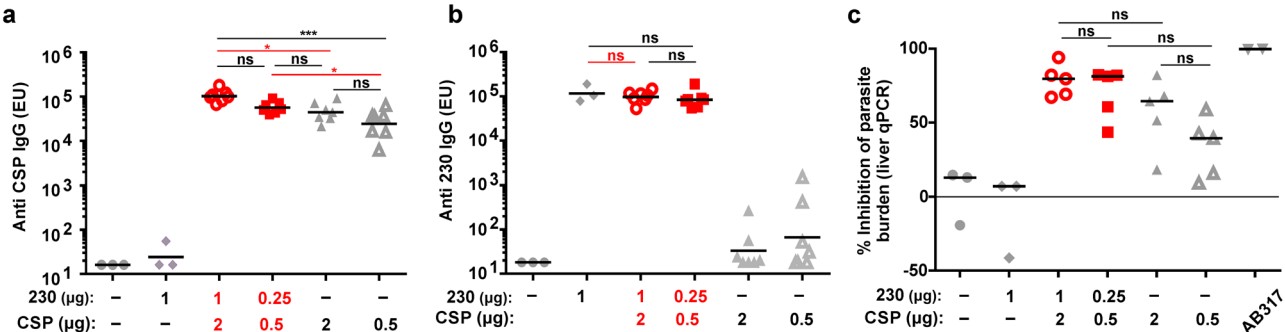

**Fig. 7 Immunization with CPQ/CSP/230 protects mice from sporozoite challenge.** C57BL/6 mice received 3 intramuscular immunizations on day 0, 14 and 28. On day 40 (**a**) anti-CSP and (**b**) anti-230 IgG EU was measured prior to challenge. **c** Mice were challenged intravenously with 2,000 PbPfCSP sporozoites and % inhibition in parasite liver burden was determined by using qPCR to assess parasite 18S rRNA relative to mouse GAPDH mRNA, compared to non-immunized mice. This experiment was performed with n = 5 mice for the CSP/230 high and low dose group, as well as the CSP high and low dose group; and n = 3 of mice for the 230 group and CPQ alone group. For **a**, **b** the line represents geometric mean, and one-way ANOVA followed by Tukey test using log-transformed data was used to analyze differences, $^{***}p < 0.0005$, $^*p < 0.05$. Only *p* values among mice immunized with homologous antigens are shown. For **c**, the line represents median, and a Kruskal-Wallis test was used to analyze differences. Red symbols or outlines indicate bivalent, liposome-displayed CSP/230.

CPQ induces functional antibodies capable of reducing sporozoite infection.

In a second rabbit study, CSP and Pfs230D1+ were used to immunize Japanese White rabbits intramuscularly on day 0, 21 and 42, with sera collection on day 0, 21, 42, 56, 84, 112, 140, and 168. Based on ELISA analysis, bivalent, liposome-displayed antigens ("CSP/230/CP") induced higher antigen-specific IgG ELISA units compared to rabbits immunized in the same conditions but with admixture with Alum on day 56 and afterwards, for both CSP (*p* = 0.018, Supplementary Fig. S10a)

and Pfs230D1+ (*p* = 0.010, Supplementary Fig. S10b). The rates of decay for both formulations over 4 months following the last vaccination were comparable (*p* = 0.704 for CSP and *p* = 0.905 for 230). Compared to immunization with monovalent CSP/CP, the bivalent CSP/230/CP post-immune sera had significantly higher CSP antibody levels (*p* < 0.001) suggesting not only a lack of immune interference, but actually an immune-enhancing effect by the addition of Pfs230D1+ component. Compared to 230/CP, CSP/230/CP did not have statistically lower Pfs230D1+ antibody levels (*p* = 0.497). Monovalent liposome vaccines did not induce

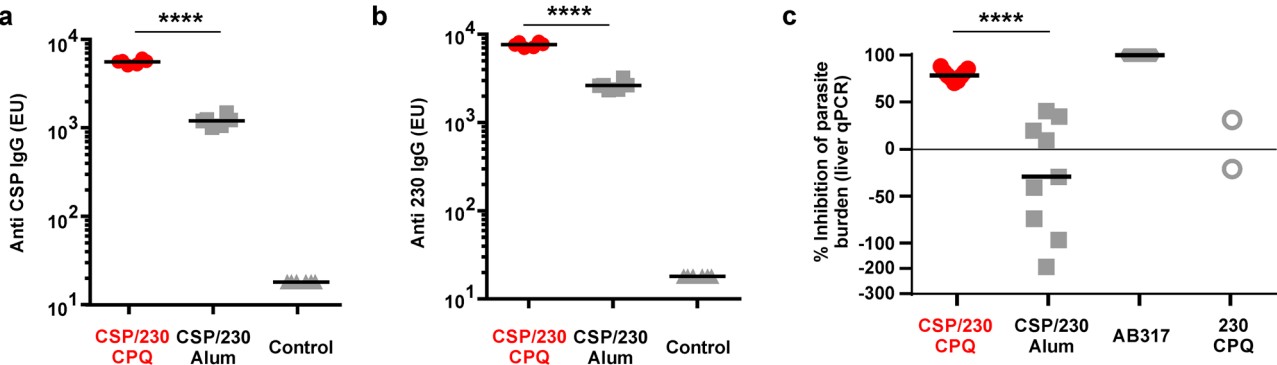

**Fig. 8 Rabbit antibodies induced by CSP/230/CPQ and passively transferred to mice confer protection to transgenic sporozoite challenge.** Rabbits were thrice immunized with CSP (20 µg) and 230 (10 µg), admixed with CPQ or Alum. 10 mg purified IgG was then intravenously injected into mice 18 h prior to sporozoite challenge with PbPfCSP, transgenic *P. berghei* parasites expressing *P. falciparum* CSP. Prior to challenge, antigen-specific rabbit antibodies in mouse sera were detected against (**a**) CSP and (**b**) 230, as determined by ELISA. Control mice did not receive any IgG. **c** Parasite liver burden (assessed with qPCR) in mice challenged with intravenous inoculation of 2000 PbPfCSP sporozoites. A human anti-CSP mAb (mAb317, 300 µg) was used as a positive control. The percent inhibition in parasite liver burden was determined by using qPCR to assess parasite 18S rRNA relative to mouse GAPDH mRNA, relative to mice that did not receive IgG. For (**a**) and (**b**), the lines represent geometric mean, and two-sided Student T-test using log-transformed data was used to analyze differences, $^{****}p < 0.0001$. For **c**, the lines represent median and a Mann-Whitney U-test was used to analyze differences, $^{****}p < 0.001$. Red symbols indicate bivalent, liposome-displayed CSP/230.

appreciable antibodies against the CSP or Pfs230D1+ antigens that were not included in the formulation. The rabbits monitored for weight over time did not exhibit weight loss, providing preliminary evidence for tolerability of CSP/230/CPQ in rabbits, a common model for vaccine toxicology (Supplementary Fig S11).

## Discussion
The feasibility of co-delivery of CSP and Pfs230D1+ using CoPoP liposomes was explored as a potential multi-stage *P. falciparum* malaria vaccine. The inclusion of two separate recombinant protein antigens into a single vaccine presents certain challenges, especially for nanoparticle-based vaccines. However, the use of CoPoP liposomes, which enables facile, non-covalent yet stable binding of conventional, His-tagged antigens simplified this process. The difference in molecular weight of CSP and Pfs230D1+ was useful in directly demonstrating their binding to liposomes based on the Ni-NTA bead competition assay, which uses SDS-PAGE as a read-out for binding efficacy and can easily resolve both proteins simultaneously. Two-color fluorescence labeling further confirmed the binding and stability of the antigens to CoPoP liposomes. Both CSP and Pfs230D1+ rapidly bound to liposomes containing CoPoP, but not liposomes lacking it. The use of two antigens co-displayed on liposomes may simplify manufacturing, compared to generating two separate sets of liposomes, each bearing an individual antigen. Future work is required to determine how a dual antigen vaccine would be characterized for clinical development. This study demonstrated that conformational monoclonal antibodies could detect Pfs230D1+ and CSP after they bound the liposomes. This is reassuring for further vaccine development, as it shows the integrity of the protein remains intact with those specific epitopes remaining accessible. Furthermore, although not demonstrated in this study, the use of these antibodies could potentially be beneficial for developing immunological methods to quantify both proteins in a vaccine formulation.

Bivalent antigen binding remained largely intact during human serum incubation in physiological conditions. Previously, we found that improved antigen delivery to immune cells in particle form compared to soluble form in the draining lymph node is a general mechanism for the efficacy of the CoPoP system[31]. We previously reported that Pfs230C1 could be taken up by immune cells in vitro more efficiently than Pfs25[32]. In the present study, a

different antigen, Pfs230D1+, was used, and it differs from Pfs230C1 (a.a. 443–731) by 109 amino acids. Pfs230D1+ addresses prior proteolytic cleavage and glycosylation of the larger molecule (Pfs230C1), and appears to potentially behave more like the hapten-like Pfs25 and thus stands to benefit more from particle presentation[32].

The intact, liposome-displayed bivalent antigens gave rise to functional antibody responses against both CSP and Pfs230D1+. Antibodies recognized native sporozoites and gametocytes, inhibited parasite development in the mosquito midgut and prevented infection in a sporozoite challenge. For anti-CSP immunity, protection could be mediated by antibodies, as mice receiving passively transferred purified rabbit IgG had reduced parasite burden following sporozoite challenge. In mouse models of malaria, the cellular response contributes to protection[49]. Indeed, it has been shown that CSP combined with liposomes including QS-21 (which enhances cellular responses) is more effective than liposomes lacking QS-21 for conferring protection in mouse sporozoite challenge models[50]. Studies have shown that mice immunized with CSP peptides induced specific CD8+ T cells that induced protection against sporozoite challenge[51,52], which reflects the importance of cellular responses in anti-infection malaria immunity[53]. Although we did not assess whether cellular responses contributed to vaccine efficacy in this study, we showed that antigen-specific polyfunctional T cell responses were induced and enhanced by QS-21 found in CPQ liposomes but lacking in CP liposomes (Fig. 6). Additional areas of study could include assessing CSP antibodies induced in varying conditions via other in vitro assays such as inhibition of liver stage development assay or traversal assays. Furthermore, passively transferring IgGs from the immunized animals into liver-humanized mice prior to the challenge of these mice with *P. falciparum* sporozoites delivered by mosquito bite would be a meaningful future direction.

When mice were immunized with nanogram doses of antigen (Fig. 4a, b and Supplementary Tables S2 and S4), the bivalent group that received half of the doses compared to the monovalent groups, showed no significant reduction in either CSP or Pfs230D1+ antibody responses. We also observed no statistically significant immune interference with the CSP antibody response nor Pfs230D1+ antibody response when mice were immunized with the same microgram dose of bivalent versus monovalent

antigens (Supplementary Fig S4 and Supplementary Table S5). Indeed, the Pfs230D1+ appeared to enhance the CSP antibody response, although further investigation is needed.

Altogether, this proof of principle study demonstrates the suitability of the CoPoP vaccine platform for inducing functional immune responses for multi-stage malaria vaccine research. Further analysis is required to determine how induced immunity would inhibit development of naturally circulating genetically diverse *P. falciparum* strains. Furthermore, future development efforts would be required to determine how such a bivalent nanoparticle vaccine could be developed for human testing. The recent advancement of the CoPoP vaccine platform into human trials for a SARS-CoV-2 vaccine (ClinicalTrials.gov Identifier: NCT04783311) may facilitate such efforts.

## Conclusion

In summary, CoPoP liposomes admixed with two malaria vaccine antigens Pfs230D1+ and CSP formed stable particles. The formulation was stable at 37 °C for over 24 h and the conformational integrity of the antigens was confirmed based on monoclonal antibody reactivity. Mice immunized with liposome-displayed bivalent antigens elicited specific antibodies that were shown to be functional against the pre-erythrocytic and sexual stage of *P. falciparum* parasites. Overall, we found CoPoP liposomes to be a suitable platform for targeting distinct stages of the malaria life cycle for vaccine development.

## Material and Methods

**Materials**. His-tagged Pfs230D1+ was produced in a baculovirus system as previously reported[30]. His-tagged $CSP_{27-383}$ was produced in *L. lactis* as previously reported[6]. CoPoP was produced as previously described[31]. The following lipids were used: 1,2-dipalmitoylsn-glycero-3-phosphocholine (DOPC, Corden # LP-R4-078, cholesterol (PhytoChol, Wilshire Technologies), Monophosphoryl Hexa-acyl Lipid A, 3-Deacyl (Synthetic) (PHAD, Avanti Cat # 699855 P) and QS-21 was obtained from Desert King. Alhydrogel 2% aluminum gel (Alum, Accurate Chemical and Scientific Corporation; Cat # A1090BS). Human conformation-dependent monoclonal antibodies 230AS-18 and 230AS-73 confer potent TRA in SMFA and were isolated from Pfs230D1M-specific memory B cells from Malian volunteers who had received vaccinations with Pfs230D1-EPA/AS01 (Clinical-trials.gov # NCT02942277). Human monoclonal antibodies Ab236 and Ab1488 were isolated from plasmablasts of RTS,S/AS01 vaccinated individuals and both are CSP C-terminal region–specific mAbs[38].

**Liposome preparation**. Liposomes were prepared by ethanol injection and nitrogen-pressurized lipid extrusion[31]. Ethanol and phosphate-buffered saline (PBS) were preheated at 50 °C. Lipids were dissolved in preheated ethanol for 10 min, followed by adding preheated PBS into the samples for another 10 min at 50 °C. Liposomes were then passed through 200, 100 and 80 nm stacked polycarbonate filters in a lipid extruder (Northern Lipids) with nitrogen pressure. After extrusion, liposomes were dialyzed against PBS to remove ethanol. Final liposome concentration was adjusting to 320 µg/mL CoPoP, and passed through a 0.2 µm, sterile filter and stored at 4 °C. The liposome formulation had a mass ratio of [DOPC:CHOL:PHAD:CoPoP] [20:5:0.4:1]. Liposome size and polydispersity were determined by dynamic light scattering with a NanoBrook 90 plus PALS instrument after 200-fold dilution in PBS.

**Particleization characterization**. Protein binding with Pfs230D1+ and CSP or the bivalent CSP/ Pfs230D1+ was carried out by incubating protein and liposomes with a mass ratio of [CSP: 230]:[CoPoP]=[1:0.5]:[1, 2, 3 or 4] for 3 h at room temperature and then the mixture was stored at 4 °C overnight. To determine the percentage of binding of total antigen to liposomes, a high-speed centrifugation assay was used to separate soluble-form antigen and particle-form antigen. Incubated samples were centrifuged at 27,000 g for 3 h at 4 °C, supernatant was collected and total antigen binding to liposomes was determined by measuring the BCA absorbance at 562 following manufacturer's protocol, using the following formula: % antigen binding = (1- $OD_{562}$ filtered liposomes + antigen/ $OD_{562}$ filtered antigen) ×100%. $OD_{562}$ is the absorption at 562 nm as measured by a microplate reader.

**Ni-NTA competition test**. To further assess protein binding, Ni-NTA Magnetic Beads (ThermoFisher # 88831) were used to compete with pre-bound proteins on the liposomes with a mass ratio of [CSP: Pfs230D1+ : CoPoP] = [1:0.5:1, 2, 3 or 4]. Sufficient beads were added to ensure full binding of the free proteins in the sample. The samples were incubated with the beads for 30 min before the supernatant and magnetic beads were separated and collected using a magnetic separator (ThermoFisher # 12321D). The beads were then resuspended in PBS. Denaturing reducing loading dye was then added to all samples (supernatant and beads) and heated to 100 °C for 10 min. The samples were then loaded into Novex 4–12% Bis-Tris acrylamide gel (Invitrogen # NP0321BOX) and subjected to PAGE and bands were visualized with Coomassie staining.

**Antigen labeling with fluorescent dyes**. Pfs230D1+ was labeled with DY-490-NHS-Ester (DY-490, Dyomics) and CSP was labeled with DY-590-NHS-Ester (DY-590, Dyomics). Labeling was carried out with DY-490 to Pfs230D1+ at molar ratio of 5:1. 150 µg of Pfs230D1+ was dialysis into 100 mM sodium bicarbonate buffer (pH 9) for 4–6 h at 4 °C twice, and then labeled with DY-490 at room temperature for 1 h. Free dyes were removed by dialysis against PBS three times at 4 °C. CSP was labeled with DY-590 in a similar manner. Labeling was carried out with DY-590 to CSP at molar ratio of 5:1. 150 µg of antigen was dialysed into 100 mM sodium bicarbonate buffer (pH 9) for 4–6 h at 4 °C twice and later labeled with DY-590 at room temperature for 1 h, followed by dialysis against PBS three times at 4 °C to remove free dyes.

**Fluorescent quenching assay**. Fluorophore-labeled Pfs230D1+ and CSP were incubated with liposomes with a 1:0.5:4 mass ratio of CSP: Pfs230D1+ : CoPoP or PoP at a final antigen concentration at 30 µg/ml. The quenching of each sample was checked at 0.5, 1 and 3 h at room temperature. To check the fluorescence signal, each of the incubation samples were diluted 200 times in PBS in a 96-well plate, and fluorescence was measured at excitation/ emission at 491/515 nm for DY-490 labeled Pfs230D1+ and excitation/ emission at 580/599 nm for DY-590 labeled CSP. The percentage binding was calculated based on the following formula: % antigen binding = (1-FL liposomes+antigen/ FL antigen) ×100%, where FL stands for fluorescent intensity.

**Serum stability**. The mixture of DY-490 labeled Pfs230D1+ (20 µg/ml) and DY-590 labeled CSP (40 µg/ml) was incubated with CoPoP liposomes (320 µg/mL CoPoP) for 3 h at room temperature. An equal volume of 40% human serum, diluted in PBS, was added to achieve a final concentration at 20% human serum. Samples were incubated at 37 °C at 0, 6, 24 and 36 h.

**Slot Blot**. Liposomal samples were formed with Pfs230D1+, CSP or CSP/ Pfs230D1+ at a CSP: Pfs230D1+ : CoPoP = 1:0.5:4 mass ratio. A 48-well slot blot apparatus (Cat # M1706545 from Bio-Rad) was set up as described in the manufacturer instructions. 50 µL of each sample was slowly applied into each well, then allowed to flow through a 0.2 µm nitrocellulose membrane (Thermo Scientific,Cat # 88013) by gravity. The membrane was removed and blocked using 5% BSA in PBS for 30 min at RT, followed by incubating with 1000× diluted monoclonal antibodies against Pfs230D1+ (230AS-18 and 230AS-73) and against CSP (Ab236 and Ab1488) for 1 h at RT. The membrane was washed with PBS for 5 min twice, then incubated with HRP anti-human IgG (cat # 109-035-098 from Jackson ImmunoResearch) for 30 min at RT. After incubation, the membrane was washed for 5 min with PBS two times. The membrane was developed using HRP substrate (VisiGlo™ HRP Chemiluminescent Cat # 97064-146) and imaged using a Bio-Rad ChemiDoc Imager.

**Murine immunization**. Three animal studies were performed at SUNY Buffalo. 5-week-old female ICR mice ($n = 10$) received intramuscular injections on days 0 and 21 containing the indicated antigens combined with CP liposomes with the following formulation, [DOPC:CHOL:PHAD:CoPoP] [20:5:0.4:1]. For the dual antigen immunization, 150 ng of total antigen contained 100 ng of CSP and 50 ng of Pfs230D1+ admixed with CP and for individual antigen immunization, 100 ng of Pfs230D1+ or 200 ng of CSP admixed with CP for 3 h at room temperature followed by incubating at 4 °C overnight. Alum was diluted with PBS to 3 mg mL$^{-1}$ concentration and then mixed with an equal volume of diluted antigen.

5-week-old female ICR mice ($n = 5$) received intramuscular injections on days 0 and 21 containing the indicated antigens combined with CP liposomes with the following formulation, [DOPC:CHOL:PHAD:CoPoP] [20:5:0.4:1]. For the dual antigen immunization, 0.75 µg or 3 µg of total antigen contained 0.5 µg of CSP and 0.25 µg of Pfs230D1+ admixed with CP for low dose injection and 2 µg of CSP and 1 µg of Pfs230D1+ admixed with CP for high dose injection and for individual antigen immunization, 1 µg/0.25 µg of Pfs230D1+ or 2 µg/0.5 µg of CSP admixed with CP for 3 h at room temperature followed by incubating at 4 °C overnight.

5-week-old female C57BL/6 mice ($n = 6$) received intramuscular injections on days 0 and 21 containing the dual antigens combined with indicating adjuvants. CP liposomes with the following formulation, [DOPC:CHOL:PHAD:CoPoP] [20:5:0.4:1], CPQ liposomes with the following formulation, [DOPC:CHOL:PHAD:-CoPoP:QS-21] [20:5:0.4:1:0.4]. The following liposomal adjuvant were used as for comparison, 2HP liposomes with the following formulation, [DOPC:CHOL:PHAD:PoP] [20:5:0.4:1] and LPQ liposomes with the following formulation, [DOPC:CHOL:-PHAD:QS-21] [20:5:1:1]. For the dual antigen immunization, 150 ng of total antigen contained 100 ng of CSP and 50 ng of Pfs230D1+ admixed with liposomal adjuvants for 3 h at room temperature followed by incubating at 4 °C overnight. Alum was diluted with PBS to 3 mg mL$^{-1}$ concentration and then mixed with an equal volume of diluted antigen.

Serum was collected on day 42 and sent to the Laboratory of Malaria and Vector Research at the National Institute of Allergy and Infectious Diseases (LMVR, NIAID) for anti-Pfs230D1+ and anti-CSP ELISA[54] and SMFA[55] analysis. ELISA plates were coated either with Pfs230D1+ or CSP proteins (100 ng/well), and each test sample (either serum or purified total IgG) was applied to the plates in triplicates at multiple dilutions. The absorbance of each test well was converted into ELISA units using a standard curve generated by serially diluting the standard (a pool of mouse

anti-Pfs230D1+ or anti-CSP antisera) in the same ELISA plate. The ELISA unit value of a standard was assigned as the reciprocal of the dilution giving an absorbance at 405 nm of 1 in a standardized assay. The SMFA was conducted with 16-18 day old gametocyte cultures of the *P. falciparum* NF54 line, and female *Anopheles stephensi* mosquitoes were fed mixtures of gametocytes and test (or control) purified total IgGs at indicated concentrations through a membrane-feeding apparatus. All feeding experiments were performed with human complement (i.e., using non-heat-inactivated human serum). The blood-fed mosquitoes were kept for 8 days and dissected ($n = 20$ per group) to enumerate the oocysts in the midgut.

**Splenocyte assay**. Splenocytes were harvested from the immunized ICR or C57BL/6 mice on day 42. Spleens were collected and then passed through a 70 µm cell strainer in a 50 mL tube to collect single cells. Cells were centrifuged at 500 relative centrifugal force (rcf), and red blood lysis buffer (Invitrogen, Cat. 00-4333-57) was added for 5 min on ice to lyse red blood cells. After incubation, 20 mL of PBS was added to dilute the lysis buffer, and samples were centrifuged at 500 rcf for 5 min. In 96-well culture plate, $1 \times 10^6$ cells/well were stimulated with 1 µg/ml of antigen and cultured in RPMI 1640 medium, with 10% FBS, 1% Penicillin-Streptomycin Solution, 1 mM pyruvate and 1 mM non-essential amino acids, 50 µM 2-Mercaptoethanol, the cells were cultured at 37 °C in a humidified atmosphere containing 5% CO$_2$. In order to check IFN-γ secretion, culture medium was collected after 48 hr, and IFN-γ secretion levels were measured based on IFN-γ mouse ELISA kit (Fisher Scientific, Cat. 50-183-06). In order to check triple-cytokine positive T cells, splenocytes were stimulated with 1 µg/mL of Pfs230D1+ or CSP for 18 h, followed by incubation with brefeldin A (Biosciences, Cat. # 555029) for another 6 h to block the cytokine secretion from the cells. Cells were stained for the surface markers using TCRβ APC/Cy7, CD4 PE/Cy7, CD8 PreCP/Cy5.5, CD44 BV605, Live/Dead marker (Cat. L34957) diluted in FASC buffer (cold-PBS containing 0.5% BSA and 0.05% sodium azide) for 25 min on ice. The cells were washed with FASC buffer twice, then fixed with the fixation/permeabilization buffer (BD cytofix/perm kit; Biosciences Cat. # 555028) for 10 min on ice. The cells were wash twice with FACS buffer, and permeabilization buffer (BD cytofix/perm kit; Biosciences Cat. # 555028) was added into each well for 20 min on ice. Intracellular markers including IFN-γ Pacific Blue, TNFα PE, Foxp3 Alex Fluor 488, IL2 PE/TexasRed were diluted in permeabilization buffer, and cells were stained for 25 min on ice. Stained cells were washed twice with permeabilization buffer, then resuspended in FASC buffer prior to BD LSRFortessa TM X-20 flow cytometry and the data were analyzed by FlowJo (version 10) software.

**Indirect immunofluorescence assay (IFA)**. *P. falciparum* gametocytes was obtained from the Johns Hopkins Parasitology Core Facility and fixed on slides as described before[31]. Slides were blocked with 5% BSA/PBS-T for 30 min at 37 °C. Sera collected from mice immunized with Pfs230D1+, CSP, or the bivalent, adjuvanted with CoPoP liposomes, were diluted 1:200 and incubated in 5% BSA/PBS with the fixed slides at 37 °C for 1 h, followed by 3 times washing with PBS in a humidity chamber, each time for 5 min. FITC-conjugated goat anti-mouse IgG (1:1000) was then incubated with the slides for 30 min at 37 °C, followed by 3 times washing with PBS, each time for 5 min. The slides were mounted with Prolong gold antifade with DAPI (# P36931) and imaged with an EVOS FL microscope (Life Technologies) using a 100× objective lens.

**New Zealand white rabbit immunization.** 10–12 week old female rabbits ($n = 2$ per group) at Pocono Rabbit Farm (Canadensis, PA, USA) received 3 intramuscular immunizations on days 0, 21 and 42 with 20 μg of CSP and 10 μg of Pfs230D1+ with CPQ or Alum adjuvant. One more rabbit was immunized with 20 μg of Pfs230D1+ with CPQ as a negative control. Serum samples were collected on day 56, pooled serum was made for each group (except Pfs230D1+ alone group), and total IgGs were purified using Protein G columns (GE Healthcare, Uppsala, Sweden) following manufacturer's instructions. The IgGs were buffer exchanged with 1× PBS, and concentrated to 100 mg/mL.

**Sporozoite preparation for challenge study.** A transgenic *P. berghei* ANKA parasite line, PbPfCSP-luc (referred to as PbPfCSP herein), was used, which expresses PfCSP instead of PbCSP under *pbcsp* promoter and GFP and luciferase constitutively[44]. Cryopreserved *P. berghei*-infected erythrocytes were injected into female ICR mice (CLEA Japan, Tokyo, Japan) to obtain asexual and sexual stage parasites. For feeding experiments, infected ICR mice were fed to *Anopheles stephensi* (SDA 500 strain) mosquitoes, and fully engorged mosquitoes were selected and kept at 20 °C until dissection. At days 10–14 post-feeding, the numbers of oocysts were examined to determine the prevalence. Sporozoites were collected from salivary glands by dissection at day 21 postblood meal in RPMI1640 medium containing 10% fetal calf serum, then diluted to 2000/100 μL in RPMI1640 medium containing 2% rat heat-inactivated serum.

**Passive immunization challenge assay.** Six-week-old female C57BL/6 mice (CLEA Japan) were intravenously injected with 10 mg/mouse of purified total IgG. As a positive control, 300 μg AB317 was injected into five mice. After 18 h, 2000 PbPfCSP sporozoites were intravenously injected into passively immunized mice. Forty-four hours later, the livers were perfused and homogenised in 5 ml of Trizol (Thermo Fisher Scientific) for total RNA extraction, followed by real-time RT-PCR analysis. The parasite burden in the liver was estimated by the levels of parasite 18 S rRNA normalized to mouse *gapdh* mRNA expression[56]. Experiments were performed using five to nine mice per group, and percent of inhibition calculated using the average amount of parasite burden from ten non-IgG-transfused mice, were plotted on the graph.

**Active immunization challenge assay.** Eight-week-old female C57Bl/6 mice (CLEA Japan) were immunized by intramuscular injection with 2/0.5 μg of CSP and 1/0.25 μg Pfs230D1+ admixed with CPQ liposomes (D0). To evaluate the interference effect of dual antigens, only CSP (2 or 0.5 μg) admixed with CPQ liposome was used as antigens in comparator groups. For control, CPQ liposomes only or Pfs230D1+ admixed with CPQ liposomes were inoculated. Immunization was repeated twice with 2 week intervals, and 12 days after the last immunization (D40), sera were collected from tail vein for the ELISA to evaluate the antibody titers of CSP and Pfs230D1+. On D42, 2,000 of PbPfCSP sporozoites were intravenously inoculated into each mouse and the liver was perfused and collected at 44 h postinoculation, as described above. For the positive control, 300 μg AB317 was injected into two naïve mice 18 h prior to sporozoite inoculation. The parasite burden and inhibition effect were determined as described above.

**Rabbit durability study using Japanese white rabbit.** Following immunization was performed at Kitayama Labs (Ina, Japan). 10-12 week old female and male rabbits received 3 intramuscular immunizations on days 0, 21 and 42 of the following conditions:

30 μg of CSP or 15 μg of Pfs230D1+ with CPQ, CPQ liposomes (40 μg CoPoP, 16 μg PHAD and 16 μg QS-21), or bivalent (30 μg CSP plus 15 μg Pfs230D1+) with CPQ liposomes (80 μg CoPoP, 32 μg PHAD and 32 μg QS-21), with a [DOPC: CHOL: PHAD: CoPoP: QS-21 = 20: 5: 0.4: 1] mass ratio. Group of animals consisted with 3 females and 3 males were immunized with similar bivalent antigen formulated in Alum adjuvant. Sera were collected and individual body weight was measured on days 0, 21,42, 56, 84, 112, 140, and 168. Serum was transferred to LMVR, NIAID for ELISA.

**Statistical analysis and reproducibility.** Sample sizes are generally indicated in the methods and figure captions. Immunization results were reproduced in outbred mice, inbred mice and rabbits. For ELISA results, one-way ANOVA followed by Tukey test using log-transformed data was used, and the same tests were utilized for cellular responses using original (nonlog-transformed) values. For SMFA, the 95% confidence interval (95%CI) and p-value were calculated using a zero-inflated negative binomial (ZINB) model as before[55]. For the rabbit durability study, ELISA units on days 56, 84, 112, 140 and 168 were compared among groups by a multifactorial linear regression model using log-transformed data. The decay rates among different groups during the same time period were evaluated by an extra sum-of-squares F test. All statistical tests were performed in JMP13 (SAS Institute, Cary, NC, USA), Prism 8 (GraphPad Software, La Jolla, CA, USA) or R (version 3.5.3, The R Foundation for Statistical Computing) and $p$ values < 0.05 are considered significant.

**Ethics.** All experiments involving mice at SUNY Buffalo were carried out using protocols approved by SUNY at Buffalo Institutional Animal Care and Use Committee. All experiments involving mice in Ehime University were carried out using protocols approved by the Institutional Animal Care and Use Committee of Ehime University, and the experiments were conducted according to the Ethical Guidelines for Animal Experiments of Ehime University. Rabbit immunization was carried out by Pocono Rabbit farm (Canadensis, PA, USA) and Kitayama Labes (Ina, Japan) according to approved protocols.

**Reporting summary.** Further information on research design is available in the Nature Research Reporting Summary linked to this article.

## Data availability
The data that support the findings of this study are available from the corresponding author upon request.

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

## Acknowledgements
We thank Dr. Fidel Zavala for providing the transgenic *P. berghei* parasites expressing PfCSP. We also thank A. Konishi and S. Sadaoka for preparing infected mosquitoes and maintaining immunized mice. This study was supported by the National Institutes of Health (R01AI148557, R01CA247771) and the Global Health Innovative Technology Fund (G2019-111), and partially supported by JSPS KAKENHI Grant (JP20H03481, JP21H02724, JP19H03459). The experiments conducted at the National Institute of Allergy and Infectious Diseases (NIAID) were supported by the Intramural Research Program of NIAID, NIH and by PATH's Malaria Vaccine Initiative.

## Author contributions

Conceptualization: W.C.H., J.L., Methodology: W.C.H., M.T.M., K.M., T.T., T.I., and J.L., Providing reagents: C.H.C., P.E.D., and Y.W., Investigation: L.Z., M.B., M.T., Mo. To., E.T., E.L., J.P., R.K., C.H.C., P.E.D., C.L., T.T., K.M., Y.W., T.I., and J.L., Supervision: J.L., Writing and editing: W.C.H., K.M., and J.L.

## Competing interests

W-C.H. and J.F.L hold interest in POP Biotechnologies. Other authors declare no competing interests.
