## [Peer Review File · Communications Biology]

Reviewers' comments:

Reviewer #1 (Remarks to the Author):

Significance:

The article by Huang et al. provides detailed analysis of CoPoP liposomes admixed with two malaria antigens Pfs230 and PfCSP for their transmission/infection blocking activity. These liposomes are stable at 37C, conformational integrity of antigens is intact and can elicit antigen specific cellular immune response in the animals. These findings represent significant advance by targeting more than one Plasmodium life cycle stages.

Comments:

Change "supporting" table# to "Supplementary" table throughout the manuscript.

Supplementary Table 3. The mean/median oocyst numbers and % oocyst prevalence per for each feed in SMFAs should be provided for clarity.

Figure 4 and Supplementary Table 3. Only 20 mosquitoes were dissected per experiment in SMFAs which are too low. If the % prevalence of infection in mosquitoes is lower, then more mosquitoes should be dissected for examining the TBA of the antibodies.

Figure 5D. DAPI signal is observed outside the parasite stages shown. Please provide better or cropped images.

Line 81-89: Appropriate references should be cited for work done on TBV candidates i.e. Pf48/45, Pfs25, Pfs28 and PfHAP2.

Reviewer #2 (Remarks to the Author):

Huang et al. report on a new potential vaccine candidate against malaria that elicits immunity against both the parasite's pre-erythrocytic and transmission stages. The vaccine combines the CSP pre-erythrocytic antigen and a fragment of the Pfs230 sexual stage antigen (Pf230D1) in immunogenic liposomes and 3-component adjuvant system. The authors show that immunization with these liposomes elicits the production of antibodies against both CSP and Pf230D1 that bind sporozoites and gametocytes, respectively, and functionally inhibit hepatic and mosquito infection, also respectively. The manuscript is clear and the conclusions are supported by the data. The results are of added value for the community and deserve publication. However, some aspects do need to be addressed beforehand, as detailed below.

- The Introduction covers the most important aspects of the topic at hand. However, I think the order in which the information is provided is not necessarily the most appropriate. Specifically, the section pertaining pre-erythrocytic vaccines goes somewhat back and forth, starting with a mention of CSP, followed by RTS,S and then R21, then explaining what the latter it is composed of, then back to explaining the domains in CSP and the immune responses it elicits, and ending with the composition of RTS,S. This seems rather illogical and a bit all over the place. Why not start by explaining what CSP is and does, detailing what immune responses it can elicit, then introduce RTS,S explaining how it is designed, followed by R21 and its design, possibly ending with the remaining CSP-based subunit vaccines mentioned in this paragraph?

- Sentence "Pfs230 is a large protein containing over 3,000 amino acids (aa), fragments of which have been assessed as candidate TBV antigens." should be appropriately referenced.

- The authors should explain more clearly the rationale for the selection of residues 552-731 of Pfs230 for the construction of Pfs230D1. Are these residues chosen taking in account genetic variation i.e. was a more conserved region chosen to incorporate on the surface of liposomes? Why do these differ from the 542-736 aa stretch in Pfs230D1M?

- "...their C-terminus his tag inserts...". Histidine is abbreviated "His", not "his", so please change to "His-tag", as in Fig. 1A.

- The data in Fig. 4 are very nice and informative as to the humoral immunogenicity of the bivalent antigens with the CoPoP liposomes. However, it would be nice to have a functional assay for the anti-CSP antibodies, similarly to what the authors did for the Pfs230 antibodies using SMFA. This is important to show that the anti-CSP antibodies elicited by immunization are indeed functional in inhibiting hepatic infection by *P. falciparum*. Such evaluation could be performed in vitro through Invasion Inhibition Assays employing HC04 cells or human primary hepatocytes, or in vivo, by passively transferring IgGs from the immunized animals into liver-humanized mice prior to challenge of these mice with *P. falciparum* sporozoites delivered by mosquito bite. The same applies to the data in Fig. 5.

- The order of the subsequent Results sections makes little sense to me. Why outbred mice for measurement of immune responses, then rabbits for passive transfer and challenge experiments, then inbred mice for challenge studies, then rabbits again for measurement of immune responses? Why don't the authors follow a logical order and describe their results on the immunogenicity in outbred mice (first the measurement of humoral and cellular responses, then the functionality by SMFA and challenge studies), then inbred mice (first the measurement of humoral and cellular responses, then the functionality by SMFA and challenge studies), and then do the same for the rabbits (first the measurement of humoral and cellular responses, then the functionality by SMFA and passive transfer + challenge studies)? As it is, in section "Antibody responses induced by bivalent, liposome-displayed CSP and Pfs230D1+ in inbred mice", the authors claim that they assessed vaccination in C57BL/6 mice prior to PbPfCSP sporozoite challenge but they do not present any data on the hepatic infection of immunized vs non-immunized mice after challenge with PbPfCSP sporozoites. In fact, it is not until section "Bivalent, liposome-displayed CSP and Pfs230D1+ induce protective immune responses" that PbPfCSP sporozoites were employed to challenge mice into which rabbit immune serum was passively transferred, and then to challenge inbred immunized mice with PbPfCSP sporozoites. The whole order in which these data are presented makes no sense to me.

- It is unclear to me why the authors initially resorted to the rabbit model and passive transfer of rabbit serum instead of simply challenging the immunized inbred mice, as they later did.

- The authors do not discuss the innovation of the development of a Pfs230D1+/PfCSP vaccine in a field context. The discussion should also address the genetic diversity of the antigens in question.

- Supplementary Fig S2 and S3 - % inhibition of TRA: The authors should consider increasing the number of mosquitoes in these experiments. As Medley et al suggested, using less than 50-100 mosquitoes might provide unreliable estimates of TRA. Alternatively, the exact same experimental conditions should be replicated at least 3 times.

- Supplementary Fig S3: days 8 later should be 8 days later.

Figure 7C, 8C: yy axis should mention which genes were involved in the assessment of the inhibition of parasite burden, otherwise, without reading the legend, it is not clear whether this was done by qPCR or IFA.

Response to Reviewer #1

The article by Huang et al. provides detailed analysis of CoPoP liposomes admixed with two malaria antigens Pfs230 and PfCSP for their transmission/infection blocking activity. These liposomes are stable at 37C, conformational integrity of antigens is intact and can elicit antigen specific cellular immune response in the animals. These finding represent significant advance by targeting more than one Plasmodium life cycle stages.

Author Response: Thank you for the encouraging assessment.

Change “supporting” table# to “Supplementary “table throughout the manuscript.

Author Response: We have done this.

Supplementary Table 3. The mean/median oocyst numbers and % oocyst prevalence per for each feed in SMFAs should be provided for clarity.

Author Response: As suggested, we have added the mean number of oocysts, the numbers of infected mosquitoes and the number of dissected mosquitoes for each test condition to Supplementary Table 3.

Figure 4 and Supplementary Table 3. Only 20 mosquitoes were dissected per experiment in SMFAs which are too low. If the % prevalence of infection in mosquitoes is lower, then more mosquitoes should be dissected for examining the TBA of the antibodies.

Author Response: We and other have shown that TBA values fluctuate depending on mean oocyst in the controls even when the same sample is tested. In other words, it is not feasible to compare TBA data accurately if samples are tested in different assays/studies with different mean control oocysts. More importantly, we have shown that TBA is determined by TRA and mean control oocysts both in SMFA¹ and DMFA². Therefore, we limit our focus to TRA in this work, by which people can meaningfully compare SMFA activity, which would not be the case for TBA.

For the number of dissected mosquitoes (n=20), we report 95%CI of TRA estimates (using mathematical modeling as described³). In this way, readers can see uncertainty of each of TRA estimate under the current test condition. In general, it is true that n=40 assay has more power to detect a small difference than n=20 assay. But the important point is that even in this relatively "underpowered" assay (n=20), we still see statistically significant inhibitions, meaning the functional activities of anti-Pfs230 antibodies were strong. For the reviewer's reference, we have evaluated the power systematically in a previous study ⁴. As shown in Fig 6 of that work, there is only minor difference in power between n=60 assay and n=20 assay as far as TRA>50%.

In addition, we note to the referee that the use of n=20 mosquitos in SMFA analytical testing is preceded in recent early-phase human clinical TBV trials.⁵⁻⁸

References:

1. Miura, K. et al. Transmission-blocking activity is determined by transmission-reducing activity and number of control oocysts in Plasmodium falciparum standard membrane-feeding assay. *Vaccine* 34, 4145-4151 (2016).
2. Miura, K. et al. Evaluation and modeling of direct membrane-feeding assay with Plasmodium vivax to support development of transmission blocking vaccines. *Sci. Rep.* 10, 12569-12569 (2020).
3. Swihart, B.J., Fay, M.P. & Miura, K. Statistical Methods for Standard Membrane-Feeding Assays to Measure Transmission Blocking or Reducing Activity in Malaria. *J Am Stat Assoc* 113, 534-545 (2018).
4. Miura, K. et al. Qualification of standard membrane-feeding assay with Plasmodium falciparum malaria and potential improvements for future assays. *PLoS One* 8, e57909 (2013).
5. Healy, S.A. et al. Pfs230 yields higher malaria transmission-blocking vaccine activity than Pfs25 in humans but not mice. *J. Clin. Invest.* 131 (2021).
6. de Graaf, H. et al. Safety and Immunogenicity of ChAd63/MVA Pfs25-IMX313 in a Phase I First-in-Human Trial. *Front. Immunol.* 12, 694759 (2021).
7. Chichester, J.A. et al. Safety and immunogenicity of a plant-produced Pfs25 virus-like particle as a transmission blocking vaccine against malaria: A Phase 1 dose-escalation study in healthy adults. *Vaccine* 36, 5865-5871 (2018).
8. Talaat, K.R. et al. Safety and Immunogenicity of Pfs25-EPA/Alhydrogel®, a Transmission Blocking Vaccine against Plasmodium falciparum: An Open Label Study in Malaria Naïve Adults. *PLoS One* 11, e0163144 (2016).

Figure 5D. DAPI signal is observed outside the parasite stages shown. Please provide better or cropped images.

Author Response: We have cropped the images as suggested.

Line 81-89: Appropriate references should be cited for work done on TBV candidates i.e. Pf48/45, Pfs25, Pfs28 and PfHAP2.

Author Response: Thank you. We have added the following passage to the introduction: "Numerous TBV antigens have been the focus of vaccine research efforts, including Pf48/45, Pfs25, Pfs28, PfHAP2 and Pfs230" [18-20].

18. Huang W-C, Sia ZR, Lovell JF: Adjuvant and Antigen Systems for Malaria Transmission-Blocking Vaccines. *Advanced Biosystems* 2018, 2:1800011.
19. Duffy PE: Transmission-Blocking Vaccines: Harnessing Herd Immunity for Malaria Elimination. *Expert Review of Vaccines* 2021, 20:185-198.
20. Singh SK, et al: A Reproducible and Scalable Process for Manufacturing a Pfs48/45 Based Plasmodium falciparum Transmission-Blocking Vaccine. *Front Immunol* 2021, 11:606266-606266.

Response to Reviewer #2

Huang et al. report on a new potential vaccine candidate against malaria that elicits immunity against both the parasite's pre-erythrocytic and transmission stages. The vaccine combines the CSP pre-erythrocytic antigen and a fragment of the Pfs230 sexual stage antigen (Pf230D1) in immunogenic liposomes and 3-component adjuvant system. The authors show that immunization with these liposomes elicits the production of antibodies against both CSP and Pf230D1 that bind sporozoites and gametocytes, respectively, and functionally inhibit hepatic and mosquito infection, also respectively. The manuscript is clear and the conclusions are supported by the data. The results are of added value for the community and deserve publication. However, some aspects do need to be addressed beforehand, as detailed below.

Author Response: Thank you for this assessment.

The Introduction covers the most important aspects of the topic at hand. However, I think the order in which the information is provided is not necessarily the most appropriate. Specifically, the section pertaining pre-erythrocytic vaccines goes somewhat back and forth, starting with a mention of CSP, followed by RTS,S and then R21, then explaining what the latter is composed of, then back to explaining the domains in CSP and the immune responses it elicits, and ending with the composition of RTS,S. This seems rather illogical and a bit all over the place. Why not start by explaining what CSP is and does, detailing what immune responses it can elicit, then introduce RTS,S explaining how it is designed, followed by R21 and its design, possibly ending with the remaining CSP-based subunit vaccines mentioned in this paragraph?

Author Response: Thank you for the guidance. As suggested, we have re-ordered the introduction as suggested above.

Sentence "Pfs230 is a large protein containing over 3,000 amino acids (aa), fragments of which have been assessed as candidate TBV antigens." should be appropriately referenced.

Author Response: We have referenced this sentence with Tachibana et al. Vaccine 37:1799-1806, 2019.

The authors should explain more clearly the rationale for the selection of residues 552–731 of Pfs230 for the construction of Pfs230D1. Are these residues chosen taking in account genetic variation i.e. was a more conserved region chosen to incorporate on the surface of liposomes? Why do these differ from the 542-736 aa stretch in Pfs230D1M?

Author Response: To try to better clarify this, we have added the following explanation:

The TBV antigen Pfs230D1+ (abbreviated as "230" herein in figures and in some descriptions for simplicity), comprises residues 552–731 of Pfs230 (NF54 allele) and was selected for optimized expression of an intact, non-glycosylated, properly folded immunogen in the baculovirus expression system ¹. This amino acid range also avoids an O-linked glycosylation site and corresponds to the cleaved-prodomain and the first cysteine motif domain of the protein

1. Lee, S.M. et al. The Pfs230 N-terminal fragment, Pfs230D1+: expression and characterization of a potential malaria transmission-blocking vaccine candidate. *Malar J* 18, 356 (2019).

- "...their C-terminus his tag inserts...". Histidine is abbreviated "His", not "his", so please change to "His-tag", as in Fig. 1A.

Author Response: These have been corrected in the text and figures.

The data in Fig. 4 are very nice and informative as to the humoral immunogenicity of the bivalent antigens with the CoPoP liposomes. However, it would be nice to have a functional assay for the anti-CSP antibodies, similarly to what the authors did for the Pfs230 antibodies using SMFA. This is important to show that the anti-CSP antibodies elicited by immunization are indeed functional in inhibiting hepatic infection by *P. falciparum*. Such evaluation could be performed in vitro through Invasion Inhibition Assays employing HC04 cells or human primary hepatocytes, or in vivo, by passively transferring IgGs from the immunized animals into liver-humanized mice prior to challenge of these mice with *P. falciparum* sporozoites delivered by mosquito bite. The same applies to the data in Fig. 5.

Author Response: Thank you for this comment. In this work, the function of anti-CSP antibodies induced by CSP/230/CoPoP combination was assessed by passive transfer of rabbit sera in the mouse challenge model (Figure 8). We agree with the reviewer that it would be ideal to test functionality of all anti-CSP antibodies generated. Unfortunately, at present we have not successfully set-up such assays in our collective laboratories. The in vivo challenge model (active and passive), which we used in this study, is a low-throughput assay (in our facility), so key conditions were tested after initial screening by ELISA. We have tried to address this point in the discussion section of the manuscript as follows:

Additional areas of study could include assessing CSP antibodies induced in varying conditions via in vitro assays such as inhibition of liver stage development assay or traversal assays. Furthermore, passively transferring IgGs from the immunized animals into liver-humanized mice prior to challenge of these mice with *P. falciparum* sporozoites delivered by mosquito bite would be a meaningful future direction.

The order of the subsequent Results sections makes little sense to me. Why outbred mice for measurement of immune responses, then rabbits for passive transfer and challenge experiments, then inbred mice for challenge studies, then rabbits again for measurement of immune responses? Why don't the authors follow a logical order and describe their results on the immunogenicity in outbred mice (first the measurement of humoral and cellular responses, then the functionality by SMFA and challenge studies), then inbred mice (first the measurement of humoral and cellular responses, then the functionality by SMFA and challenge studies), and then do the same for the rabbits (first the measurement of humoral and cellular responses, then the functionality by SMFA and passive transfer + challenge studies)? As it is, in section "Antibody responses induced by bivalent, liposome-displayed CSP and Pfs230D1+ in inbred mice", the authors claim that they assessed vaccination in C57BL/6 mice prior to PbPfCSP sporozoite challenge but they do not present any data on the hepatic infection of immunized vs non-immunized mice after challenge with PbPfCSP sporozoites. In fact, it is not until section "Bivalent, liposome-displayed CSP and Pfs230D1+ induce protective immune responses" that PbPfCSP sporozoites were employed to challenge mice into which rabbit immune serum was passively transferred, and then to challenge inbred immunized mice with PbPfCSP sporozoites. The whole order in which these data are presented makes no sense to me.

Author Response: Thank you again for the constructive suggestion. We have rearranged the results to follow this more logical order (inbred mice; outbred mice; rabbits)

It is unclear to me why the authors initially resorted to the rabbit model and passive transfer of rabbit serum instead of simply challenging the immunized inbred mice, as they later did.

Author Response: Our rationale for doing this was to confirm that functional antibodies were being induced (since cellular immunity may play a role in protection in the murine challenge model). However, we agree that the data presentation in the revised manuscript flows better.

The authors do not discuss the innovation of the development of a Pfs230D1+/PfCSP vaccine in a field context. The discussion should also address the genetic diversity of the antigens in question.

Author Response: We have briefly acknowledged this in the discussion as follows:

Altogether, this proof of principle study demonstrates the suitability of the CoPoP vaccine platform for inducing functional immune responses for multi-stage malaria vaccine research. Further analysis is required to determine how induced antibodies would inhibit development of naturally circulating genetically diverse *P. falciparum* strains. Furthermore, future development efforts would be required to determine how a bivalent nanoparticle vaccine could be developed for human testing. The recent advancement of the CoPoP vaccine platform into human trials

for a SARS-CoV-2 vaccine (ClinicalTrials.gov Identifier: NCT04783311) may facilitate such efforts.

Supplementary Fig S2 and S3 - % inhibition of TRA: The authors should consider increasing the number of mosquitoes in these experiments. As Medley et al suggested, using less than 50-100 mosquitoes might provide unreliable estimates of TRA. Alternatively, the exact same experimental conditions should be replicated at least 3 times.

Author Response: While the Medley paper suggested 50-100 mosquitoes in the abstract, we do not believe there is evidence to support that specific numbers (ie why 20 are not enough, etc.) based on that paper. While that may have been a general recommendation at the time it was published thirty years ago, studies done within last 10 years¹⁻³ have shown that the uncertainty of TRA estimates are not only determined by the number of dissected mosquitoes, but also mean oocyst number in the control group and level of TRA. Importantly, we include the 95% confidence interval for TRA estimates (the details of mathematical modeling have been described⁴), which allow readers to see uncertainty in the estimates under given test conditions. We believe this is a reasonable approach. For the reviewer's reference, we previously evaluated the impact of dissection number on the uncertainty of TRA estimate systematically in previous study⁴. As shown in Fig 6, of that work there is only minor difference in power between n=60 assay and n=20 assay as far as TRA>50%. Finally, we note to the referee that the use of n=20 mosquitos in SMFA analytical testing is preceded in recent early-phase human clinical TBV trials.⁵⁻⁸

References:

1. Miura, K. et al. Qualification of standard membrane-feeding assay with Plasmodium falciparum malaria and potential improvements for future assays. PLoS One 8, e57909 (2013).
2. Churcher, T.S. et al. Measuring the blockade of malaria transmission--an analysis of the Standard Membrane Feeding Assay. Int. J. Parasitol. 42, 1037-1044 (2012).
3. Miura, K. et al. An inter-laboratory comparison of standard membrane-feeding assays for evaluation of malaria transmission-blocking vaccines. Malar. J. 15, 463 (2016).
4. Swihart, B.J., Fay, M.P. & Miura, K. Statistical Methods for Standard Membrane-Feeding Assays to Measure Transmission Blocking or Reducing Activity in Malaria. J Am Stat Assoc 113, 534-545 (2018).
5. Healy, S.A. et al. Pfs230 yields higher malaria transmission-blocking vaccine activity than Pfs25 in humans but not mice. J. Clin. Invest. 131 (2021).
6. de Graaf, H. et al. Safety and Immunogenicity of ChAd63/MVA Pfs25-IMX313 in a Phase I First-in-Human Trial. Front. Immunol. 12, 694759 (2021).
7. Chichester, J.A. et al. Safety and immunogenicity of a plant-produced Pfs25 virus-like particle as a transmission blocking vaccine against malaria: A Phase 1 dose-escalation study in healthy adults. Vaccine 36, 5865-5871 (2018).
8. Talaat, K.R. et al. Safety and Immunogenicity of Pfs25-EPA/Alhydrogel®, a Transmission Blocking Vaccine against Plasmodium falciparum: An Open Label Study in Malaria Naïve Adults. PLoS One 11, e0163144 (2016).

Supplementary Fig S3: days 8 later should be 8 days later. Figure 7C, 8C: yy axis should mention which genes were involved in the assessment of

the inhibition of parasite burden, otherwise, without reading the legend, it is not clear whether this was done by qPCR or IFA.

Author Response: Thank you again for your careful reading. We corrected the Fig S3 caption. For Figure 7C and 8C, we have revised the axis legend to indicate the measurement is liver qPCR based and we revised the figure caption to indicate inhibition was determined by using qPCR to assess parasite 18S rRNA relative to mouse GAPDH mRNA, compared to non-immunized control mice.

REVIEWERS' COMMENTS:

Reviewer #1 (Remarks to the Author):

The authors have satisfactorily addressed my concerns. I look forward to this manuscript getting published at Communications Biology.

Reviewer #2 (Remarks to the Author):

The authors have addressed my concerns satisfactorily.